# Continual Learning with Gated Incremental Memories for Sequential Data Processing

## Abstract

The ability to learn over changing task distributions without forgetting previous knowledge, also known as continual learning, is a key enabler for scalable and trustworthy deployments of adaptive solutions. While the importance of continual learning is largely acknowledged in machine vision and reinforcement learning problems, this is mostly under-documented for sequence processing tasks. This work focuses on characterizing and quantitatively assessing the impact of catastrophic forgetting and task interference when dealing with sequential data in recurrent neural networks. We also introduce a general architecture, named Gated Incremental Memory, for augmenting recurrent models with continual learning skills, whose effectiveness is demonstrated through the benchmarks introduced in this paper.

## 1 Introduction

Continual Learning (CL) can be defined as *"the unending process of learning new things on top of what has already been learned"* (Ring, 2011).
A more formal definition centered around the computational aspects of a CL algorithm is presented in Lesort et al. (2019):

**Definition 1** *Given a potentially infinite sequence of unknown distributions $\mathcal{D} = \{D_1, D_2, D_3, ...\}$, where each $D_i = \{X_i, Y_i\}$ includes the input data $X_i$ and the target labels $Y_i$, a Continual Learning algorithm $A_i^{CL}$ is defined by the following signature:*

$$\forall D_i \in \mathcal{D}, \quad A_i^{CL} \equiv \; < h_{i-1}, \, TR_i, \, M_{i-1}, \, t_i > \; \rightarrow \; < h_i, \, M_i >$$

*where $TR_i$ is the training set drawn from the corresponding data distribution $D_i$, $h_i$ is the function that the model has learned after having seen all the training sets up to the i-th, $M_i$ is an external memory in which to store patterns from $TR_i$ and $t_i$ is the label identifying the task.*

In this paper, the notion of *task* is associated to a specific objective (e.g. learn to classify MNIST digits). An *input distribution*, instead, generates data related to a particular task (e.g. subsets of MNIST digits). Hence, each task is associated to multiple input distributions (also called subtasks). In the CL scenario, a learning model is required to incrementally build and dynamically update internal representations as the distribution of tasks varies across its lifetime. Ideally, part of such internal representations will be general and invariant enough to be reusable across similar tasks, while another part should preserve and encode task specific representations. Unfortunately, continuous plasticity of internal representations under drifting task distributions is widely known to suffer from negative interference between the tasks that are incrementally presented to the model, yielding to the well known stability-plasticity dilemma (Parisi et al., 2019; Zhou et al., 2012) of connectionist models. The result of this being models which catastrophically forget previously acquired knowledge (Robins, 1995; French, 1999) as new tasks become available, that is one of the main challenges faced by CL algorithms.
While current trends in CL put particular emphasis on computer vision applications or reinforcement learning scenarios, sequential data processing is rarely taken into consideration (see Section 2). Sequential data processing in CL involves a stream $\mathcal{D}$ of distributions of sequences $D_i = (X_i, Y_i), i = 1, 2, 3, ...$, where $X_i$ are sequences of vectors and $Y_i$ are the corresponding targets. In Machine Learning (ML), sequential data processing plays a fundamental role in important fields like Natural Language Processing (NLP), signal processing, bioinformatics and many

others. In this context, Recurrent Neural Networks (RNNs) provide an effective means to deal with sequential data, thanks to their adaptive memory, that is the ability of developing neural representations that capture the history of their inputs. Learning proper memory representations is a major challenge for RNNs. Especially in a CL setting, where RNNs have also to deal with drifts in task distributions which can greatly affect their capability of developing robust and effective memory representations.

In this work, we provide a threefold contribution to the discussion concerning CL in sequential data processing. First, we introduce benchmarks for the evaluation of CL skills by adapting three sequential datasets to CL scenarios. These benchmarks can be easily tuned to varying degrees of complexity and hardness, and they have been devised to provide an assessment that is not tailored to a specific application domain (e.g. NLP, RL or computer vision). Rather, they aim at evaluating the intrinsic effectiveness of recurrent models in a CL setting. Second, we define a new dynamic approach that imbues RNN architectures with CL skills by incrementally adding new modules to capture the shift in task distribution while avoiding catastrophic forgetting. This architecture, named Gated Incremental Memory, leverages gating autoencoders to automatically recognize input distributions in order to avoid any explicit supervision concerning their identity at inference time. Third, we evaluate baseline RNN models on the proposed benchmarks, comparing their performance against our enhanced architecture. Source code and full details regarding the experimental setup are provided as supplemental material, to ease replicability of the results and allow other researchers a direct and fair comparison of their own CL models on our benchmarks.

The results of our empirical analysis confirm the advantages of using our enhanced architecture over standard recurrent models. Such advantages are particularly clear when testing enhanced architecture on old distributions, since it successfully prevents forgetting, when learning sequences with increasing length (in the Copy Task) and when the input sequences end with a common suffix (in the Sequence Classification SSMNIST Task). These results highlight some of the key differences between feedforward and recurrent CL techniques, and pinpoint the need for solutions specifically designed for recurrent architectures.

## 2 RELATED WORK

CL literature mostly focuses on computer vision and reinforcement learning applications, with approaches ranging from regularization methods (Kirkpatrick et al., 2017; Zenke et al., 2017), to dual models (Hinton & Plaut, 1987; French, 1997), to dynamic architectures (Rusu et al., 2016; Yoon et al., 2018). The work of Coop & Arel (2013) is one of the first attempts to apply RNNs in CL scenarios. In their paper, they employed a RNN with Fixed Expansion Layer (FEL) (Coop & Arel, 2013) and trained the resulting model to reconstruct auto-associative binary sequences. The popular Copy task (Graves et al., 2014) and Sequential Stroke MNIST dataset (de Jong, 2016) have been used by (Sodhani et al., 2018) to test the capabilities of dynamic recurrent models in presence of increasingly longer sequences. By using dynamic external memories, Asghar et al. (2018) successfully applied RNNs to CL on NLP tasks with MNLI dataset. Finally, Kemker et al. (2018) applied feedforward networks to the Audioset dataset (Gemmeke et al., 2017) in a CL setting. Our dynamic RNN architecture is inspired by Progressive networks (Rusu et al., 2016), a CL approach popular in the feedforward domain that deals with new distributions by dynamically adding new hidden units to a feedforward layer and connecting them to the existing hidden units through lateral connections. The gating autoencoders are adapted to sequential scenarios from Aljundi et al. (2017), where they are first introduced for feedforward models.

## 3 GATED INCREMENTAL MEMORIES FOR CONTINUAL LEARNING WITH RECURRENT NEURAL NETWORKS

In this section, we introduce Gated Incremental Memory (GIM), a novel CL architecture designed for recurrent neural models and sequential data. In particular, we show how GIM can be obtained by combining together a recurrent version of the Progressive network (Rusu et al., 2016) and a set of gating autoencoders (Aljundi et al., 2017) in order to avoid, at test time, any explicit supervision about subtask label. We denote the recurrent version of the Progressive network as Augmented models (A-LMN and A-LSTM), while the entire architecture, including both augmented architectures and gating autoencoders, is called GIM. To highlight the generality of our approach, we

detail the application of GIM to two substantially different recurrent neural models from literature (GIM-LMN, GIM-LSTM). In the following, we indicate an entire sequence with bold notation (e.g. $\mathbf{x}$), while a single vector with plain formatting (e.g. $x_i$).

## 3.1 RECURRENT NEURAL NETWORKS

The proposed approach is independent on the underlying recurrent architecture. In this work, we focus our study on two different classes of RNNs, using either gated and non-gated approaches. Gated models, like LSTM (Hochreiter & Schmidhuber, 1997) and GRU (Cho et al., 2014), leverage adaptive gates to enable selective memory cell updates. In our analysis, we consider LSTM as a representative of gated architectures, given its popularity in literature and its state-of-the-art performances in several sequential data processing benchmarks. Non-gated approaches rely on different mechanisms to solve the vanishing gradient problem, like parameterizing recurrent connections with an orthogonal matrix (Mhammedi et al., 2017). In our analysis, we consider the Linear Memory Network (LMN) (Bacciu et al., 2019) as a representative of non-gated approaches. LMNs leverage a conceptual separation between a nonlinear feedforward mapping computing the hidden state $h_t$ and a linear dynamic memory computing the history dependent state $h_t^m$. Briefly, in formulas:

$$h_t = \sigma(W^{xh}x_t + W^{mh}h_{t-1}^m)$$
$$h_t^m = W^{hm}h_t + W^{mm}h_{t-1}^m$$

where $x_t$, $h_t$ and $h_t^m$ are the input vector, the functional component activation and the memory state at time $t$. The memory state $h_t^m$ is the final output of the layer, which is given as input to subsequent layers. It is then useful to study how both models behave in CL environments and how their different memory representations respond to phenomena like drifting tasks distribution, eventually resulting in catastrophic forgetting.

## 3.2 GATED INCREMENTAL MEMORY (GIM)

GIM represents a general class of dynamic, recurrent architectures that can be built on top of any recurrent model. GIM relies on a progressive memory (Rusu et al., 2016) extension of the underlying RNN model that, for the sake of this work, are LSTMs or LMNs. We also leverage a set of gating autoencoders, one for each distribution, to automatically select the module which has the best match to the current input distribution. Figures 1 and 2 provide an overview of the entire GIM architecture during training and test. The main component of GIM is the RNN *module*. As soon as a new distribution arrives, a new RNN module is added on top of the existing architecture and connected to the previous one. The exact inter-modules connections are slightly different depending on the underlying recurrent model. When using the A-LSTM, at each timestep, the new module takes as additional input the current hidden state of the previous module. In the A-LMN, the hidden state $h^t$ of the new module is computed by taking as additional input the concatenation of both previous module's memory $h_t^m$ and functional activation $h_t$. In order to prevent forgetting, when a new module is attached to the existing architecture, the previous module's parameters are frozen and no longer updated. Therefore, each module becomes an expert of its own domain. The input vector $x_t$ at each time step $t$ is forwarded to all modules. Each module has its own output layer and, during training, the last module added to the network is used to generate the final output $y$. The forward pass for an A-LMN with $N$ modules can then be formalized as follows:

$$h_{T,1}^m, h_{T,1} = \text{LMN}_1(\mathbf{x}, h_{0,1}^m)$$
$$h_{T,j}^m, h_{T,j} = \text{LMN}_j([\mathbf{x}; h_{T,j-1}^m; h_{T,j-1}], h_{0,j}^m), j = 2, ..., N$$

where $h_{T,i}^m$ and $h_{T,i}$ are the memory states and functional activations of module $i$ after having seen the entire input sequence $\mathbf{x}$ of length $T$ and $[\cdot; \cdot]$ is the concatenation operator between vectors. The aggregated output $y$ is computed by passing the final memory state $h_{T,N}^m$ through a linear layer. The A-LSTM forward pass follows the same process, substituting the final hidden state $h_T$ to the

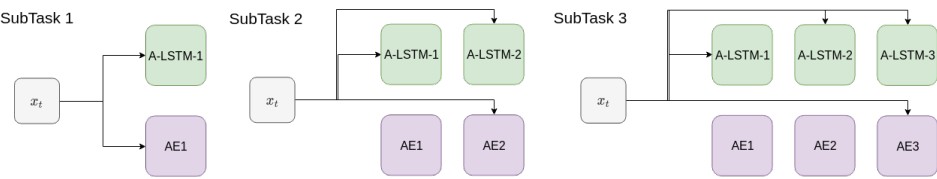

Figure 1: Training of GIM-LSTM on 3 subtasks.

memory state and functional activations of A-LMN.

Each module of the augmented architecture is associated to a single LSTM autoencoder (Srivastava et al., 2015), which is a sequence-to-sequence model trained jointly to the corresponding module to reconstruct the input $\mathbf{x}$. The training process involves only the autoencoder associated to the last added module. The loss function used to train the autoencoders is the reconstruction error, that is the mean squared error (MSE) applied element-wise to the input and reconstructed sequence. The Augmented models are trained by using standard Cross Entropy loss for classification tasks.
The following equations provide a formal description of the GIM-LMN training when tackling the $i$-th distribution (the GIM-LSTM behaves in the same way):

$$y = \text{A-LMN}(\mathbf{x})$$
$$\tilde{\mathbf{x}} = \text{AE}_i(\mathbf{x})$$

where $\text{AE}_i$ identifies the autoencoder associated to module $i$. The input $\mathbf{x}$ is sampled from the $i$-th distribution. The output $y$ and the reconstruction $\tilde{\mathbf{x}}$ are then used in the corresponding loss functions (Cross Entropy and MSE, respectively) to train the A-LMN and the autoencoder by gradient descent.
At inference time, the input $\mathbf{x}$ is forwarded to all autoencoders and the module associated to the one producing the minimum reconstruction error is selected to produce the final output, as described by the following equations:

$$\tilde{\mathbf{x}}_i = \text{AE}_i(\mathbf{x}), i = 1, ..., N$$
$$k = \underset{i}{\arg\min} \, \text{MSE}(\tilde{\mathbf{x}}_i, \mathbf{x})$$
$$y = \text{LMN}_k(\mathbf{x}).$$

The algorithms describing the forward pass for both A-LMN and A-LSTM and the training and test procedure of GIM are provided in Appendix A.1.
GIM, like Progressive networks, is capable of learning multiple distributions without being affected by forgetting. In addition, GIM overcomes one of the major drawbacks of Progressive networks (Rusu et al., 2016): it does not require explicit knowledge about input distributions at test time, since gating autoencoders are able to autonomously recognize current input and use the appropriate module for output. GIM also simplifies the inter-modules connections: while in Progressive networks they are feedforward networks, created between a module and *all* the next ones, GIM employs a simpler, non adaptive, version and connects only adjacent modules. Therefore, the connections do not contribute to the total number of adaptive parameters, which scales linearly with the number of modules. Finally, as described in Aljundi et al. (2017), it is possible to compute a relatedness measure between subtasks using the autoencoders reconstruction errors. Given a new subtask $T_k$, an old subtask $T_a$ and their associated autoencoders, the relatedness is computed by using the reconstruction errors $E_{r_k}$ and $E_{r_a}$ of the autoencoders measured on validation data for subtask $T_k$. Formally:

$$\text{Rel}(T_k, T_a) = 1 - \left( \frac{E_{r_a} - E_{r_k}}{E_{r_a}} \right). \tag{1}$$

Notice that this definition is not entirely equivalent to the one reported in Aljundi et al. (2017). Since we assume that each autoencoder achieves the lowest error on its own training data, we have $E_{r_a} > E_{r_k}$. Aljundi et al. (2017) normalize relatedness by having $E_{r_k}$ in the denominator of (1), in place of $E_{r_a}$. In our experiments, we noticed that this led to large negative values for the relatedness, which are not particularly insightful. Therefore, we propose a different normalization coefficient, using the largest reconstruction error $E_{r_a}$ in place of $E_{r_k}$.

## 4 EXPERIMENTS

One of the main contributions of this paper is the definition of a set of learning tasks that can be used as standard benchmarks for CL algorithms on sequence processing problems. We believe that the current literature is lacking a set of simple, application-agnostic, benchmarks that can provide a fair comparison and evaluation of different approaches with a minimal experimental setup. The final appendices provide all the details regarding the experimental setup needed to reproduce the results. The experiments rely on three datasets: the Copy dataset (Graves et al., 2014), the Sequential Stroke MNIST dataset

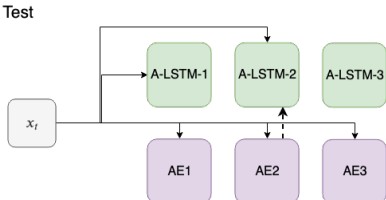

Figure 2: Test of GIM-LSTM. Input x is presented to all autoencoders. In this case, AE2 obtains the minimum reconstructione error. Hence, the second module (dashed line) is chosen to produce the output. The input is propagated up until the second module.

(de Jong, 2016) and the Audioset dataset (Gemmeke et al., 2017). We adapted each dataset to classification tasks in CL environments. We split each task in multiple subtasks, associating to each one a subset of the possible classes. The complexity of each task can be adjusted and controlled by the experimenter. The tasks are not designed for a specific application domain, but instead, they are general enough to stress the main difficulties of RNN training and CL scenarios. This is particularly useful to test the limitation of each model and allows a great degree of flexibility. The source code is available online and we plan to release more datasets in the future[1].

### 4.1 COPY TASK

The Copy task (Graves et al., 2014) consists in the reconstruction of sequences of random binary vectors. This task could be considered a simple CL example, since the data distribution for each subtask only differs for the sequences' length. However, generalizing to long sequence lengths is a difficult problem for traditional recurrent architectures (Graves et al., 2014) and therefore the same techniques used in more standard CL settings can be used

Table 1: Accuracy (in percentage) on Copy task, measured at the end of each subtask training. Column labels indicate sequence length for each subtask.

| % | 5 | 12 | 19 | 26 | 33 | 40 |
|---|---|---|---|---|---|---|
| **A-LMN** | 100 | 100 | 100 | 100 | 95 | 95 |
| **A-LSTM** | 100 | 100 | 100 | 95 | 90 | 85 |
| **LMN** | 100 | 100 | 100 | 78 | – | – |
| **LSTM** | 100 | 97 | 85 | 78 | – | – |

to solve this problem by incrementally increasing the memory capacity of the final model (Sodhani et al., 2018). Furthermore, it is useful to evaluate the memory capacity of the Augmented architecture compared against the standard RNNs. In the Copy Task, LMN and A-LMN use orthogonal initialization of the memory component (see Appendix A.2 for more details). Gating autoencoders have not been employed in this task, since its objective is exactly the reconstruction of the input sequence. The subtasks are organized in a curriculum (Bengio et al., 2009) following sequence length, from sequences with $5 \pm 2$ vectors to sequences with $26 \pm 2$, steps of 7 vectors (4 subtasks total). Results, summarized in Table 1, show that LMN performs slightly better than LSTM (using more hidden units can bridge this gap). This is expected since orthogonal models typically outperform LSTMs on pure memorization problems (Vorontsov et al., 2017). Augmented models outperformed both of them, with A-LMN converging faster than A-LSTM. Augmented models have been trained also with longer sequences (up to length 40), without sensible drops in accuracy. All models obtained an accuracy on par with a random classifier when tested on previous subtasks (a well-known problem for RNNs). The Copy Task highlights the behavior of recurrent CL models trained on growing sequence lengths and proves their effectiveness when compared to standard (non CL oriented) recurrent models. The use of dynamic modules in A-LMN and A-LSTM positively contributes to the memory capacity, justifying the additional cost required by the Augmented architectures.

---

[1]Public link redacted to preserve anonymity, reviewers can find source code in the supplementary material

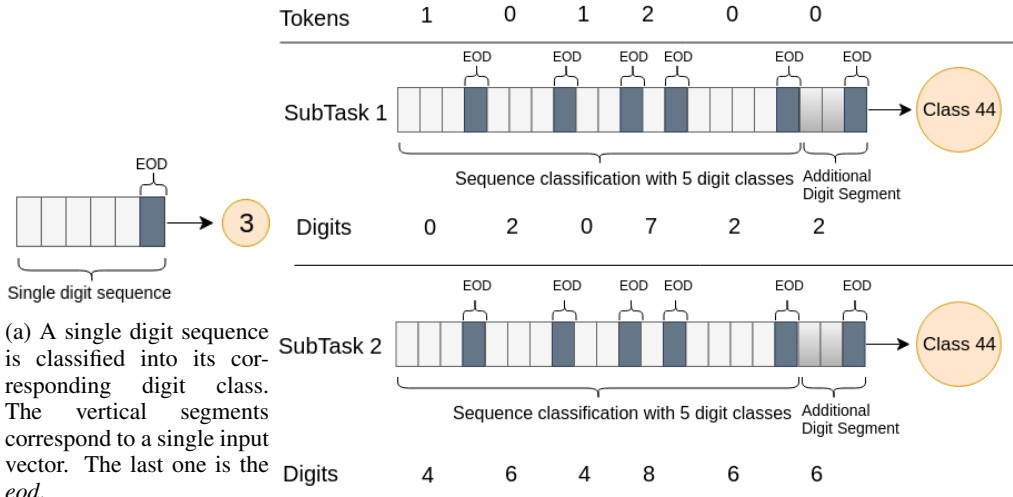

(a) A single digit sequence is classified into its corresponding digit class. The vertical segments correspond to a single input vector. The last one is the *eod*.

(b) A sequence of digits (including an additional final segment), is classified into its corresponding class. The tokens are uniquely associated to the class index 44. The digits corresponding to the tokens are explicitly showed for each subtask. The last token is common to all the sequences belonging to the same subtask and its length affects the learning behavior, as showed in Table 3.

Figure 4: SSMNIST Tasks description

## 4.2 SSMNIST TASK

Sequential Stroke MNIST (SSMNIST) (de Jong, 2016) is a variation of the popular MNIST dataset of handwritten digits. In this version, each digit is composed by a sequence of pen strokes, where each pen stroke is represented by two coordinates ($x$ and $y$, scalars), an end of stroke bit *eos* and an end of digit bit *eod*. We devised two different kind of experiments, aimed at both evaluating gating autoencoders capabilities and GIMs classification performances.

In Sodhani et al. (2018), the SSMNIST dataset has already been employed in CL environments. However, the authors build the CL strategy by progressively increasing the sequence length (e.g. by concatenating together multiple digits). Here, we chose a different approach by limiting each subtask to a fixed set of digit classes. Our approach is consistent with the literature about CL in computer vision, where the different subtasks are designed over different object classes within the images (Maltoni & Lomonaco, 2019; Lopez-Paz & Ranzato, 2017).

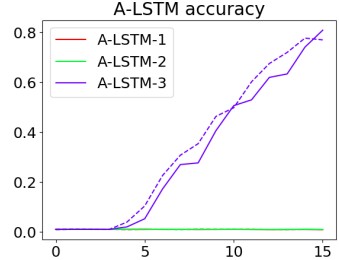

Figure 3: A-LSTM learning curve on Sequence Classification. Final concatenated length of 12.

### 4.2.1 SINGLE DIGIT SSMNIST

The Single Digit task (Fig. 4a), requires the models to predict the digit class associated to the input sequence. The task has been partitioned into 5 subtasks, each of which selects input patterns belonging to 2 digit classes. There is no intersection in the digit classes between different subtasks. Classification performance easily reached top (100%) accuracy for all models. The only exception is the LMN, which obtained an accuracy on par with a random classifier on subtask 3. LMN and LSTM experienced complete forgetting of old knowledge and were not able to address any of the previous subtasks. GIMs, instead, completely prevent forgetting through the use of gating autoencoders, that were always able to distinguish between the 5 subtasks at test time. There was no clear difference between GIM-LSTM and GIM-LMN performances in this task. GIMs are useful in this context not

Table 3: The table reports the values of the final digit segment length that produce, for each model, the behavior specified by the top row. *L* (Learning) means that the model learns all the subtasks, *NL* (No Learning) means that the model does not learn any subtask and *PL* (Partial Learning) means that the model learns only one subtask.

| | L | NL | PL |
|---|---|---|---|
| **GIM-LMN** | $\leq 22$ | $-$ | 35 |
| **GIM-LSTM** | $\leq 11$ | $\geq 14$ | $12 - 13$ |
| **LMN** | 16 | $\geq 24$ | 22 |
| **LSTM** | $\leq 11$ | $\geq 12$ | $-$ |

only to achieve top accuracy on all subtasks (e.g. GIM-LMN against LMN on subtask 3), but also to prevent forgetting. Final test performances are summarized in Table 2.

### 4.2.2 SEQUENCE CLASSIFICATION SSMNIST

The Sequence Classification task (Fig. 4b) partitions the 10 digit classes in 3 subtasks of 3 digit classes each, without intersections (see Table 4).

Sequences are composed by 5 tokens (either 0, 1 or 2), each one associated to a specific digit, depending on the subtask. The association between tokens and digits for each subtask is described by Table 4. Sequences are grouped into 100 different classes, according to the

Table 2: Accuracies (in percentage) on test sets after training on all subtasks on Single Digit SSMNIST.

| % | T1 | T2 | T3 | T4 | T5 |
|---|---|---|---|---|---|
| **GIM-LMN** | 100 | 100 | 100 | 100 | 100 |
| **GIM-LSTM** | 100 | 100 | 100 | 100 | 100 |
| **LMN** | 0 | 0 | 0 | 0 | 100 |
| **LSTM** | 0 | 0 | 0 | 0 | 100 |

sequence of tokens they represent. As an example, class 44 consists of *tokens* $(1, 0, 1, 2, 0)$. The complete list of classes is presented in Appendix A.4.

The models are trained to classify each sequence into its corresponding class. This task is more difficult than the Single Digit, since the models have first to decode the digits sequence from the input sequence (e.g. $(4, 6, 4, 8, 6)$), then associate such sequence to the appropriate class (44) by inferring the correct tokens, without any intermediate help.

Gating autoencoders were still able to recognize the subtask, even if, with respect to the Single Digit, this task increases the number of digit classes per subtask (from 2 to 3) and it is based on longer sequences (made by multiple digit sequences). As in the Single Digit, GIMs successfully overcame catastrophic forgetting, while LMN and LSTM were not able to address previous subtasks.

By adding a common digit segment at the end of each sequence, the classification problem becomes much more difficult. The common suffix between all sequences requires a larger memory capacity to distinguish the inputs. Depending on the length of the final segment, different behaviors emerged, as showed in Table 3. Some lengths completely prevented learning (*No Learning* column), while some others did not affect the performances (*Learning* column), still obtaining an accuracy above $95\%$. The most interesting behavior (*Partial Learning* column) has been observed for a narrow range of lengths. In this case, learning is prevented on the first 2 subtasks. On the third subtask, the models suddenly achieved the top accuracy, as showed in Fig. 3.

The Partial Learning behavior highlights how the use of GIMs enables learning of more articulated input distributions, compared to baseline RNNs.

### 4.3 AUDIOSET TASK

Audioset (Gemmeke et al., 2017) is a collection of annotated audio events, extracted from 10 seconds audio clips and organized hierarchically in classes. The objective is the classification of a sound from its audio clip source, embedded through a VGG-acoustic model into 10 vectors, one per second. In order to implement a CL scenario, we selected 40 audio classes and split them among 4 subtasks (10 classes per subtask). We selected the 40 classes according to the procedure outlined by Kemker et al. (2018). Since their classes have not been published, we randomly selected them from the superset resulting from their preprocessing pipeline. See Appendix A.5 for a complete description. Results showed that gating autoencoders were perfectly capable of detecting the

correct data distributions at inference time. Therefore, GIMs successfully prevented forgetting: they obtained a final test performance comparable to the validation accuracy measured at the end of each subtask training. Standard RNN architectures, on the contrary, were prone to catastrophic forgetting, achieving $0\%$ accuracy on previous subtasks. Table 5 summarizes these results.

Audioset data has already been used in literature to assess CL skills Kemker et al. (2018). However, the work by Kemker et al. (2018) focused on the task from a *static* perspective, relying on the use of feedforward models only. Since the preprocessing step provides, for each audio clip, a sequence of fixed-size embeddings, it is possible to concatenate the vectors into a single large vector and feed it to the network. The sequential aspect of the task, however, is completely lost. At the best of our knowledge, we are the first to tackle Audioset in CL scenarios with recurrent models. It is also important to notice that the task difficulty is increased when using recurrent networks, since the model is not able to see the input in its entirety (like in feedforward networks), but it has to scan it one timestep at a time.

## 4.4 RELATEDNESS ANALYSIS

In our experiments, large relatedness values are associated to "heterogeneous" distributions. We call heterogeneous distributions the ones that generate patterns from many different classes (as in Audioset) or from a complex mixture of few classes (as in Sequence Classification SSMNIST). We observed that, in correspondence with heterogeneous distributions, autoencoders training converges to larger values of reconstruction error than the ones obtained with homogeneous distributions (as in Single Digit SSMNIST). However, reconstruc-

| Tokens-to-Digits | 0 | 1 | 2 |
|:---:|:---:|:---:|:---:|
| **T1** | 2 | 0 | 7 |
| **T2** | 6 | 4 | 8 |
| **T3** | 3 | 1 | 5 |

Table 4: This table associates to each subtask of Sequence Classification SSMNIST (T1, T2, T3) the values taken by the tokens (0, 1, 2) in the sequence classes. As an example, sequence $(1, 0, 1, 2, 0)$ (class 44) corresponds to different multi digit sequences: $(0, 2, 0, 7, 2)$ for subtask 1 and $(4, 6, 4, 8, 6)$ for subtask 2.

tion error, hence relatedness, does not tell anything about gating capabilities (i.e. the ability to select the appropriate module at test time). Autoencoders, in fact, always performed correctly in all experiments, preventing forgetting. A complete account of relatedness measure is provided in Appendix A.3.

## 5 CONCLUSIONS

The main objective of this work is to draw attention on the problem of CL for sequential data processing. We proposed a set of benchmarks against which future models can assess their performances. These benchmarks test the main difficulties of the sequential and CL domains and their difficulty can be carefully tuned. We introduced GIM, a recurrent CL architecture for sequential data processing, and showed that it con-

Table 5: Accuracies (in percentage) on test sets after training on all subtasks on Audioset.

| % | T1 | T2 | T3 | T4 |
|:---:|:---:|:---:|:---:|:---:|
| **GIM-LMN** | 61 | 65 | 53 | 50 |
| **GIM-LSTM** | 60 | 69 | 60 | 47 |
| **LMN** | 0 | 0 | 0 | 43 |
| **LSTM** | 0 | 0 | 0 | 42 |

sistently outperforms baseline RNNs on all the proposed benchmarks. The results support the claim that recurrent architectures need to be adapted to manage CL scenarios and encourage future works towards an in-depth study of their behaviors.

In the future, GIM could be applied to Reinforcement Learning tasks, where there are more opportunities to compare performances with existing works. In order to provide a formal characterization of the learning process, it would be useful to apply existing metrics to better describe the type of tasks addressed by the model, in terms both of transfer learning opportunities and subtasks interference. New metrics can also be developed, specifically tailored to sequential learning problems.

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

## A  APPENDIX

### A.1  GATED INCREMENTAL MEMORIES

Algorithms 1 and 2 outline the forward pass for A-LSTM and A-LMN architectures.
Algorithm 3 describes the training and test process for the GIM architecture.

---

**Algorithm 1** A-LSTM Forward Pass

---

**Require:** A-LSTM with $N$ modules, input $\mathbf{x}$
**Ensure:** $N > 1$
 1: Initialize hidden state of module 1: $h_{1,i}$
 2: $h_{1,f} = \text{A-LSTM}_1(\mathbf{x}, h_{1,i})$                    ▷ Compute final state of module 1
 3: **for** $d \leftarrow 2, N$ **do**
 4:     Initialize hidden state of module $d$: $h_{d,i}$
 5:     $\hat{\mathbf{x}} = [\mathbf{x}; h_{d-1,f}]$                    ▷ Concatenate hidden state with input
 6:     $h_{d,f} = \text{A-LSTM}_d(\hat{\mathbf{x}}, h_{d,i})$                    ▷ Obtain final state of module $d$
 7: **end for**
 8: Output final hidden state of last module: $h_{N,f}$

---

---

**Algorithm 2** A-LMN Forward Pass

---

**Require:** A-LMN with $N$ modules, input $\mathbf{x}$
**Ensure:** $N > 1$
1: Initialize memory state of module 1: $h_{1,i}^m$
2: $h_{1,f}^m, h_{1,f} = \text{A-LMN}_1(\mathbf{x}, h_{1,i}^m)$     $\triangleright$ Compute final memory and functional state of module 1
3: **for** $d \leftarrow 2, N$ **do**
4:     Initialize memory state of module $d$: $h_{d,i}^m$
5:     $\hat{\mathbf{x}} = [\mathbf{x}; h_{d-1,f}^m; h_{d-1,f}]$     $\triangleright$ Concatenate functional and memory state with input
6:     $h_{d,f}^m, h_{d,f} = \text{A-LMN}_1(\hat{\mathbf{x}}, h_{d,i}^m)$   $\triangleright$ Compute final memory and functional state of module $d$
7: **end for**
8: $y_N^m = \sigma(W_N^{mo} h_{N,f}^m)$     $\triangleright$ Output of the final module

---

**Algorithm 3** Training and testing of LSTM Autoencoders, including output selection from Augmented architecture

---

**Require:** A sequence of distributions $\mathcal{D} = \{D_1, D_2, D_3, ...\}$
**Ensure:** $|\mathcal{D}| > 1$
1: $l_{ae} \leftarrow []$     $\triangleright$ Create empty list
2: **while** a new distribution $D_k$ is available **do**
3:     $\text{LSTM}_{enc}, \text{LSTM}_{dec} = \text{init-autoencoder}()$
4:     $l_{ae}.\text{append}((\text{LSTM}_{enc}, \text{LSTM}_{dec}))$     $\triangleright$ Append new autoencoder
5:     Initialize the hidden state of the encoder $h_{enc,i}$
6:     **for** training batch $\mathbf{x} \in D_k$ **do**
7:         $h_{enc,f} = LSTM_{enc}(\mathbf{x}, h_{enc,i})$     $\triangleright$ Compute encoder state
8:         $h_{dec,f} = LSTM_{dec}(\mathbf{0}, h_{enc,f})$     $\triangleright$ Compute decoder state
9:         $\tilde{\mathbf{x}} = W_f h_{dec,f}$     $\triangleright$ Compute output
10:        $J = \text{MSE}(\mathbf{x}, \tilde{\mathbf{x}})$     $\triangleright$ Compute reconstruction loss
11:        Take an optimization step using $\frac{\partial J}{\partial w}$ computed with backpropagation
12:     **end for**
13: **end while**

14: **for** test batch $\mathbf{x} \in D$ **do**
15:     $l_{rec} \leftarrow []$     $\triangleright$ Create empty list
16:     **for** $\text{LSTM}_{enc}\text{LSTM}_{dec} \in l_{ae}$ **do**
17:         $\tilde{\mathbf{x}} \leftarrow$ output of current autoencoder on $\mathbf{x}$
18:         $l_{rec}.append(\text{MSE}(\mathbf{x}, \tilde{\mathbf{x}}))$     $\triangleright$ Append the reconstruction loss
19:     **end for**
20:     $m = \arg\min l_{rec}$     $\triangleright$ Retrieve minimum loss autoencoder's index
21:     Select the output produced by the $m$-th module of the Augmented architecture
22: **end for**

---

## A.2 EXPERIMENT SETTINGS

### A.2.1 COPY

The experiment settings used for the Copy task are listed below:

1. Each vector in the sequence has $8$ binary values. The last bit is always $0$, except for the delimiter vector which is a vector of zeros with the last bit set to $1$.

2. The subtasks draw sequences with lengths from $5 \pm 2$ to $26 \pm 2$ with steps of $2$.

3. The sequence length for each training step is drawn at random according to the lengths corresponding to the current subtask.

4. Binary Cross Entropy (BCE) loss, RMSProp optimizer with learning rate of $3e - 5$, momentum of $0.9$ and no weight decay. The gradient clipping is set to $5.0$. The mini batch size is $3$.

|    | T1 | T2 | T3 | T4 | T5 |
|----|----|----|----|----|----|
| **C1** | 2  | 0  | 4  | 6  | 8  |
| **C2** | 3  | 1  | 5  | 7  | 9  |

Table 6: This table associates to the five subtasks (T) of Single Digit SSMNIST, the corresponding digit classes (C) (2 classes per subtask).

5. The LSTM and GIM-LSTM use 256 hidden units, while the LMN and the GIM-LMN use 126 memory units and 126 feedforward units. With this setting the models are better comparable since they use the same number of units in total.

6. For what concerns the LMN and the GIM-LMN, the memory matrices have been initialized as orthogonal, due to the benefits on the norms of the matrices explained in Vorontsov et al. (2017). Moreover, in order to constrain the memory matrices to remain approximately orthogonal during training, the BCE loss has been augmented with an additional penalty term in order to penalize non-orthogonal memory matrices ($\lambda \parallel WW^T - I \parallel$). The hyperparameter $\lambda$ associated with this penalty term has been set to $0.1$.

We tried to use a different number of hidden units in all models. Such changes affect the final result only for LSTM and LMN with $512$ hidden units, leading to a better performance ($90\%$ accuracy) on sequences of length $19$ and $26$. With this configuration, LMN and LSTM obtain comparable performances. We tried configuration with $256$ and $512$ hidden units for both functional and memory component for LMN and A-LMN, and with $128$ and $512$ for LSTM and A-LSTM. We varied the hyperparameter $\lambda$ in the range $[0.01, 0.1, 0.2, 1]$.

### A.2.2 SSMNIST

In the Single Digit task, subtasks are numbered from $1$ to $5$ (following the order of the training phase) and are associated to the digit classes according to Table 6.

The experiment settings for all SSMNIST tasks are listed below:

- LSTM and GIM-LSTM with $64$ hidden units and $1$ hidden layer
- LMN and GIM-LMN with $64$ hidden units both for the memory component and the functional component
- RMSProp optimizer with learning rate of $3e-5$, L2 regularizer of $1e-3$, gradient clipping with norm $5.0$ and momentum of $0.9$
- Mini-batch size of $4$ sequences
- The LSTM autoencoders use $40$ hidden units for both the encoder and the decoder in the Single Digit SSMNIST, and $200$ hidden units for the Sequence Classification SSMNIST. They are optimized with Adam optimizer, learning rate of $1e-4$ and L2 decay of $1e-3$.
- We use $60\%$ of dataset for training, $20\%$ for both validation and test.

For what concerns Sequence Classification SSMNIST Task during training, sequences are dynamically generated at each step by randomly selecting a batch of class labels and then by randomly generating sequences belonging to those classes. Each digit in the sequence is randomly sampled from the training set. During validation and final test, we generate $5000$ sequences with the same procedure. The classes are equally balanced in the validation and test sets ($50$ patterns per class).
The average sequence length in SSMNIST is $40$ vectors per digit, with $4$ scalars in each vector. We compact the representation in the LSTM autoencoders by a factor of $4$, by using $40$ hidden units per digit. Hence, in Sequence Classification SSMNIST Task we use $40 * 5 = 200$ hidden units for autoencoders.

### A.2.3 AUDIOSET

The experiment settings are listed below:

Table 7: Relatedness values between subtasks on Single Digit SSMNIST.

|    | T1     | T2     | T3     | T4     | T5     |
|----|--------|--------|--------|--------|--------|
| T1 | 1.0000 |        |        |        |        |
| T2 | 0.1703 | 1.0000 |        |        |        |
| T3 | 0.5407 | 0.4762 | 1.0000 |        |        |
| T4 | 0.3642 | 0.2876 | 0.4403 | 1.0000 |        |
| T5 | 0.0102 | 0.4432 | 0.7379 | 0.5164 | 1.0000 |

Table 8: Relatedness values between subtasks on Sequence Classification SSMNIST.

|    | T1     | T2     | T3     |
|----|--------|--------|--------|
| T1 | 1.0000 |        |        |
| T2 | 0.9899 | 1.0000 |        |
| T3 | 0.9867 | 0.9668 | 1.0000 |

- Standard LSTM and GIM-LSTM with 32 hidden units and 1 hidden layer
- LMN and GIM-LMN with 32 hidden units both for the memory component and the functional component
- RMSProp optimizer with learning rate of $3e-5$, L2 regularizer of $1e-3$, gradient clipping with norm $5.0$ and momentum of $0.9$
- Mini-batch size of $4$ audio clips
- The LSTM autoencoders use $200$ hidden units for both the encoder and the decoder and they are optimized with Adam optimizer, a learning rate of $3e-5$ and no decay.

We tested configurations with different number of hidden units, in the range $[8, 16, 32, 64]$ for all models. In the LMN and GIM-LMN the same number of hidden units has been used in both functional and memory component. Results showed that even models with 8 hidden units achieve accuracies that are only slightly below the ones achieved by the models with 32 units. By increasing the units to 16, we obtained the same results reported in the paper.

Each Audioset pattern is composed by 10 vectors with 128 elements, for a total number of elements equal to 1280. We use 200 hidden units in the LSTM autoencoders, thus compressing the representation by a factor of 6.4.

### A.3 RELATEDNESS BETWEEN SUBTASKS

The relatedness values between subtask are showed by Table 7, 8, 9.

### A.4 SSMNIST CLASSES

Table 10 reports the sequence classes used in the SSMNIST Sequential classification task.

### A.5 AUDIOSET CLASSES

The input patterns used in the experiments have been restricted to the ones labeled with only 1 class. Each class has no restrictions based on the associated ontology, is marked with a reliability $> 70\%$

Table 9: Relatedness values between subtasks on Audioset.

|    | T1     | T2     | T3     | T4     |
|----|--------|--------|--------|--------|
| T1 | 1.0000 |        |        |        |
| T2 | 0.8415 | 1.0000 |        |        |
| T3 | 0.8713 | 0.9673 | 1.0000 |        |
| T4 | 0.9126 | 0.9444 | 0.9541 | 1.0000 |

Table 10: SSMNIST classes

| ID | 1 | 2 | 3 | 4 | 5 |
|----|---|---|---|---|---|
| 1 | 0 | 2 | 1 | 2 | 1 |
| 2 | 0 | 0 | 2 | 0 | 0 |
| 3 | 2 | 1 | 0 | 2 | 2 |
| 4 | 0 | 0 | 0 | 0 | 2 |
| 5 | 1 | 1 | 1 | 1 | 0 |
| 6 | 0 | 0 | 0 | 1 | 1 |
| 7 | 1 | 2 | 1 | 0 | 2 |
| 8 | 2 | 2 | 0 | 2 | 2 |
| 9 | 1 | 2 | 2 | 2 | 1 |
| 10 | 0 | 2 | 0 | 0 | 1 |
| 11 | 0 | 0 | 1 | 2 | 2 |
| 12 | 0 | 0 | 1 | 1 | 0 |
| 13 | 1 | 1 | 1 | 0 | 1 |
| 14 | 0 | 0 | 0 | 1 | 2 |
| 15 | 1 | 2 | 2 | 0 | 2 |
| 16 | 1 | 1 | 2 | 0 | 0 |
| 17 | 1 | 2 | 1 | 0 | 0 |
| 18 | 1 | 2 | 1 | 2 | 0 |
| 19 | 0 | 1 | 1 | 0 | 0 |
| 20 | 1 | 0 | 0 | 0 | 2 |
| 21 | 1 | 0 | 1 | 1 | 1 |
| 22 | 1 | 0 | 0 | 0 | 0 |
| 23 | 2 | 1 | 1 | 1 | 0 |
| 24 | 0 | 0 | 2 | 1 | 2 |
| 25 | 2 | 1 | 1 | 2 | 0 |
| 26 | 0 | 1 | 2 | 2 | 0 |
| 27 | 2 | 1 | 1 | 2 | 2 |
| 28 | 0 | 2 | 1 | 1 | 2 |
| 29 | 2 | 0 | 0 | 2 | 2 |
| 30 | 1 | 0 | 0 | 0 | 1 |
| 31 | 0 | 1 | 1 | 1 | 2 |
| 32 | 1 | 0 | 2 | 1 | 2 |
| 33 | 1 | 2 | 2 | 1 | 0 |

| ID | 1 | 2 | 3 | 4 | 5 |
|----|---|---|---|---|---|
| 34 | 0 | 0 | 2 | 1 | 0 |
| 35 | 1 | 2 | 2 | 1 | 1 |
| 36 | 0 | 0 | 2 | 2 | 1 |
| 37 | 0 | 2 | 1 | 1 | 0 |
| 38 | 0 | 2 | 1 | 0 | 1 |
| 39 | 0 | 0 | 2 | 2 | 0 |
| 40 | 0 | 2 | 2 | 0 | 1 |
| 41 | 1 | 0 | 2 | 0 | 2 |
| 42 | 0 | 1 | 0 | 2 | 2 |
| 43 | 1 | 2 | 1 | 2 | 1 |
| 44 | 1 | 0 | 1 | 2 | 0 |
| 45 | 1 | 1 | 2 | 0 | 2 |
| 46 | 0 | 1 | 0 | 2 | 1 |
| 47 | 0 | 2 | 0 | 2 | 2 |
| 48 | 1 | 0 | 2 | 2 | 2 |
| 49 | 2 | 0 | 2 | 2 | 0 |
| 50 | 0 | 1 | 2 | 2 | 1 |
| 51 | 0 | 2 | 0 | 2 | 0 |
| 52 | 1 | 0 | 2 | 2 | 0 |
| 53 | 2 | 2 | 0 | 1 | 2 |
| 54 | 1 | 0 | 0 | 2 | 2 |
| 55 | 0 | 2 | 2 | 2 | 2 |
| 56 | 0 | 1 | 1 | 0 | 1 |
| 57 | 2 | 2 | 2 | 2 | 1 |
| 58 | 1 | 2 | 0 | 2 | 1 |
| 59 | 2 | 1 | 2 | 2 | 1 |
| 60 | 2 | 2 | 0 | 0 | 2 |
| 61 | 2 | 2 | 1 | 0 | 2 |
| 62 | 2 | 2 | 1 | 2 | 0 |
| 63 | 2 | 1 | 2 | 1 | 2 |
| 64 | 0 | 0 | 0 | 2 | 2 |
| 65 | 1 | 1 | 2 | 2 | 2 |
| 66 | 1 | 0 | 0 | 2 | 0 |

| ID | 1 | 2 | 3 | 4 | 5 |
|----|---|---|---|---|---|
| 67 | 2 | 0 | 2 | 0 | 0 |
| 68 | 1 | 2 | 1 | 1 | 2 |
| 69 | 2 | 2 | 2 | 0 | 1 |
| 70 | 0 | 1 | 0 | 0 | 2 |
| 71 | 0 | 1 | 2 | 1 | 1 |
| 72 | 1 | 1 | 0 | 2 | 0 |
| 73 | 1 | 1 | 0 | 1 | 1 |
| 74 | 2 | 1 | 1 | 2 | 1 |
| 75 | 1 | 2 | 0 | 2 | 0 |
| 76 | 1 | 0 | 1 | 2 | 2 |
| 77 | 0 | 0 | 0 | 2 | 0 |
| 78 | 0 | 2 | 0 | 0 | 0 |
| 79 | 1 | 2 | 0 | 1 | 2 |
| 80 | 2 | 1 | 0 | 1 | 1 |
| 81 | 0 | 1 | 0 | 0 | 1 |
| 82 | 1 | 1 | 1 | 2 | 2 |
| 83 | 2 | 0 | 1 | 0 | 0 |
| 84 | 1 | 1 | 2 | 1 | 0 |
| 85 | 1 | 1 | 1 | 0 | 2 |
| 86 | 0 | 2 | 1 | 0 | 2 |
| 87 | 1 | 2 | 2 | 0 | 0 |
| 88 | 0 | 1 | 0 | 1 | 1 |
| 89 | 1 | 0 | 1 | 1 | 0 |
| 90 | 2 | 1 | 2 | 0 | 1 |
| 91 | 0 | 2 | 2 | 0 | 0 |
| 92 | 0 | 1 | 1 | 1 | 0 |
| 93 | 1 | 0 | 0 | 1 | 1 |
| 94 | 2 | 0 | 0 | 1 | 1 |
| 95 | 0 | 0 | 2 | 0 | 1 |
| 96 | 2 | 2 | 0 | 1 | 0 |
| 97 | 1 | 1 | 0 | 2 | 1 |
| 98 | 2 | 0 | 2 | 1 | 0 |
| 99 | 2 | 1 | 2 | 0 | 0 |
| 100 | 0 | 0 | 1 | 1 | 2 |

and is not sub or super class of any other, following the procedure adopted by Kemker et al. (2018). Table 11 reports the 40 classes (N column) selected from Audioset dataset.

The ID is the corresponding class code and Category is its description, as in the official Audioset documentation. Train and Test list the number of patterns of the corresponding class in the entire dataset.

The first 10 (N from 1 to 10) classes are used in the first subtask, the classes from 11 to 20, from 21 to 30 and from 31 to 40 are used in the second, third and fourth subtasks.

Table 11: Audioset classes

| N | ID | Train | Test | Category |
|---|---|---|---|---|
| 1 | 4 | 154 | 6 | Conversation |
| 2 | 6 | 175 | 12 | Babbling |
| 3 | 10 | 75 | 3 | Whoop |
| 4 | 525 | 476 | 10 | Radio |
| 5 | 14 | 296 | 16 | Screaming |
| 6 | 15 | 436 | 8 | Whispering |
| 7 | 17 | 72 | 4 | Baby laughter |
| 8 | 18 | 124 | 7 | Giggle |
| 9 | 21 | 95 | 3 | Chuckle chortle |
| 10 | 23 | 483 | 14 | Baby cry infant cry |
| 11 | 368 | 393 | 11 | Microwave oven |
| 12 | 369 | 836 | 58 | Blender |
| 13 | 377 | 1396 | 59 | Vacuum cleaner |
| 14 | 382 | 201 | 11 | Electric shaver electric razor |
| 15 | 386 | 318 | 4 | Computer keyboard |
| 16 | 387 | 271 | 18 | Writing |
| 17 | 390 | 80 | 14 | Telephone bell ringing |
| 18 | 391 | 243 | 9 | Ringtone |
| 19 | 398 | 332 | 21 | Buzzer |
| 20 | 400 | 284 | 13 | Fire alarm |
| 21 | 403 | 175 | 18 | Steam whistle |

| N | ID | Train | Test | Category |
|---|---|---|---|---|
| 22 | 407 | 337 | 15 | Tick |
| 23 | 411 | 732 | 16 | Sewing machine |
| 24 | 414 | 64 | 14 | Cash register |
| 25 | 415 | 2005 | 41 | Printer |
| 26 | 419 | 382 | 48 | Hammer |
| 27 | 421 | 85 | 3 | Sawing |
| 28 | 442 | 146 | 12 | Chink clink |
| 29 | 450 | 382 | 12 | Trickle dribble |
| 30 | 452 | 94 | 5 | Fill (with liquid) |
| 31 | 453 | 1056 | 18 | Spray |
| 32 | 454 | 163 | 6 | Pump (liquid) |
| 33 | 467 | 283 | 52 | Slap smack |
| 34 | 468 | 1294 | 37 | Whack thwack |
| 35 | 469 | 339 | 42 | Smash crash |
| 36 | 470 | 99 | 10 | Breaking |
| 37 | 471 | 37 | 20 | Bouncing |
| 38 | 481 | 652 | 13 | Beep bleep |
| 39 | 483 | 384 | 17 | Ding |
| 40 | 493 | 103 | 18 | Rumble |

