# OpenReview forum: "Continual Learning with Gated Incremental Memories for Sequential Data Processing"
_ICLR.cc/2020/Conference — Reject_

### Official Review · AnonReviewer3 · 2019-10-22
**Official Blind Review #3**

**Rating:** 1

**Review:**

Summary:

In this paper, the authors propose a new method to apply continual learning on sequential data. The model is constructed by combining an Autoencoder and LSTM/LMN for each task. The experiments on several datasets show the proposed model outperforms basic LSTM/LMN.


Strength:

+ Sequential data widely exist in the real world, e.g., text, health records. Thus, It is interesting to see that continual learning is used in sequential data.

+ The motivation of the proposed model is clear. The authors save the learned knowledge in the hidden representation of LSTM/LMN.

Weakness:
- In this paper, the model size linearly increases since the number of LSTM/LMN and AE increases when a new task comes in. Thus, if the number of tasks is too large, the model size is quite big. In traditional continual learning settings, researchers may not always increase the model size for overcoming catastrophic forgetting. For example, if task 1 and task 2 sample from the same distribution, they can share the same LSTM/LMN and AE. Thus, it would be better if the authors can consider how to reduce the model size in the future version.

- In the experiments, the authors only compare the proposed model with simple LSTM or LMN. However, most continual learning methods can still be applied in this scenario, at least regularization based methods [1,2] can be simply applied in this scenario. The authors may need to compare the proposed method with them in the future version.

- It is better to compare it with a larger dataset. For example, in the natural language processing field, we can regard sentiment analysis on one language as one task. Then, we can construct the continual learning dataset for sentiment analysis.

Minor Comments:
It is better to improve Figure 3 by adding the x-axis label and y-axis label.


[1] Kirkpatrick, James, et al. "Overcoming catastrophic forgetting in neural networks." Proceedings of the national academy of sciences 114.13 (2017): 3521-3526.
[2] Zenke, Friedemann, Ben Poole, and Surya Ganguli. "Continual learning through synaptic intelligence." Proceedings of the 34th International Conference on Machine Learning-Volume 70. JMLR. org, 2017.

**Experience Assessment:**

I have published one or two papers in this area.

**Review Assessment: Checking Correctness Of Derivations And Theory:**

N/A

**Review Assessment: Checking Correctness Of Experiments:**

I carefully checked the experiments.

**Review Assessment: Thoroughness In Paper Reading:**

I read the paper thoroughly.

---

> ### Author Response · Authors · 2019-11-12
> **Response to R3**
>
> We agree that the model size is the main limitation GIM. However, as you say, it is reasonable to assume that the number of hidden units of each module could decrease as the number of modules increases. Inter-modules connections can foster reuse of previous features, thus reducing the need to learn them from scratch. As far as we know, our work is the first example of a progressive model + gating AE applied to sequential data. Therefore, we decided to leave this extension out of the proposed paper, since we prefer to focus more on the benchmark design and the sequential nature of the data rather than the model architecture. We would like to propose GIM as a simple baseline that can be improved upon in several different directions.
>
> We agree that it would be valuable to compare current CL techniques (e.g EWC) with GIM in order to further assess our approach. We will integrate some of the main techniques into our analysis and we will highlight the main differences between their application on a computer vision scenario and a sequential data processing scenario.
>
> We struggle to understand what the use of a larger dataset (e.g. NLP / sentiment analysis) would add to our analysis. We recognize that it will be an important step in the development of full-fledged CL systems for sequential data processing, but we believe that in this first phase it would be better to focus on task-agnostic benchmarks before moving on to more complex scenarios. This choice allows us to highlight the CL properties of the model without the need to tailor it on a specific field of application, which often requires a complex preprocessing or the use of intermediate embeddings. By using simpler benchmarks, we reduce the probability to misinterpretation of results. This approach also makes it easier for other researchers to compare their results against our benchmark, since we do not require a large computational infrastructure to manage the experiments.

---

### Official Review · AnonReviewer2 · 2019-10-23
**Official Blind Review #2**

**Rating:** 3

**Review:**

The goal of this work is to best understand the performance and benchmarking of continual learning algorithms when applied to sequential data processing problems like language or sequence data sets. The contributions of the paper are 3 fold - new benchmarks for CL with sequential data for RNN processing, new architecture introduced for more effective processing and a thorough empirical evaluation.

Introduction:
I think a little more insight into why the sequential data processing CL scenario is any different than the vision scenario would be quite helpful. Specifically, it would be quite impactful to tell us more about what the additional challenges with RNNs for CL vs feedforward for CL are in the intro.

The paper is written as if the benchmark is the main contribution and the architecture improvement is just a delta on top of this, but it gets confusing when the methods section starts off with just directly stating the new architecture.

The algorithm seems like a straightforward combination of recurrent progressive nets and gated autoencoders for CL. Can the authors provide more justification if that is the contribution or there is more to the insight than has been previously suggested in prior work?

Figure 1 has a very uninformative caption. It also doesn’t show how modules feed into one another properly.

The motivation for why one needs GIM after one already has A-LSTM or A-LMN is not very clear?

Overall the contribution does seem a bit incremental based on prior work and the description lacks enough detail to properly indicate why this is a very important contribution?

Experiments:
What does it mean to be application agnostic but restricted to particular datasets and losses? This doesn’t quite parse to me.

The description of the tasks is very informal and hard to follow. It’s not clear what exactly the tasks and datasets look like

“using morehidden units can bridge this gap” -> why not just do it? Its a benchmark after all.

Overall the task descriptions should be in a separate section where the setup is described in a lot of detail and motivated properly.

The results in the experiments section are very hard to parse. The captions need much more detail for eg Table 2.

Could we also possibly have more baselines from continual learning? For instance EWC (Kirkpatrick) or generative replay might be competitive baselines.

Overall I think that the GIM and A-LMN and A-LSTM methods are reasonable although somewhat incremental. But the proposed benchmarks are pretty unclear and the results are a bit hard to really interpret well. It would also be important to run comparisons with more baselines and to provide more ablation/analysis experiments to really see the benefit of GIM/A-LMN or A-LSTM. I also think that the task descriptions should be much earlier in the paper and desribed in much more rigorous detail.


**Experience Assessment:**

I have read many papers in this area.

**Review Assessment: Checking Correctness Of Derivations And Theory:**

N/A

**Review Assessment: Checking Correctness Of Experiments:**

I assessed the sensibility of the experiments.

**Review Assessment: Thoroughness In Paper Reading:**

I read the paper at least twice and used my best judgement in assessing the paper.

---

> ### Author Response · Authors · 2019-11-12
> **Response to R2**
>
> The main difference between the sequential data processing scenario and the vision scenario is related to the fact that sequential processing requires the use of a memory that embeds the history of past inputs. Such memories have to be appropriately learned and preserved, making the sequential processing tasks clearly different than the vision tasks. When it comes to CL, drifts in the input distribution could affect the hidden memory of RNNs. Additional works will be needed in order to clarify this phenomenon. We will clarify this point in the Introduction.
>
> The main concern of this work was to provide a set of common benchmarks for CL in sequential domains that are independent of domain-specific applications (e.g. NLP) against which existing and future models can compare their performances. We will better describe the experimental settings, reserving a specific section to the description of the tasks and datasets. Since they are the main contribution of this work, we agree that they should be better highlighted.
> In addition, we extend the progressive approach and the gating autoencoder to the recurrent domain. At the best of our knowledge, no previous work proposed these two extensions for recurrent neural networks, nor they combine both into one end-to-end model.
>
> The reason why the Augmented models need the autoencoders (e.g. from A-LSTM to GIM-LSTM) is that without the autoencoders is not possible to avoid the use of task labels at inference time. GIM architectures can detect the correct module for inference, while Augmented modules alone only allow the transfer of useful, learned features from one module to the others.
>
> The experimental protocol is task-agnostic in the sense that we do not restrict the choice of datasets to a particular application (e.g. NLP), but instead, we proposed a set of general datasets (Copy and SSMNIST) that do not require any domain-specific technique. Using this benchmarks, we can evaluate CL models while eliminating the idiosyncrasies of specific application domains.
>
> In future versions, we will extend standard CL techniques to the RNN scenario and we will compare their performances against both naive RNNs and GIM.
> We will also provide ablation studies highlighting the effects that autoencoders and inter-modules connections have on the overall performance of GIM.

---

### Official Review · AnonReviewer1 · 2019-10-23
**Official Blind Review #1**

**Rating:** 3

**Review:**

The paper proposed an interesting continual learning approach for sequential data processing with recurrent neural network architecture.
The authors provide a general application on sequential data for continual learning, and show their proposed model outperforms baseline.

It is natural that their naive baseline shows poor performance since they do not consider any continual learning issues like the catastrophic forgetting problem. Then, I hesitate to evaluate the model in terms of performance. In that sense, it would be much crucial to show more meaningful ablation studies and analysis for proposed model. However, there is a few of thing about them.

Then, I decide to give a lower score that even the authors suggest that the main contribution is a definition of problem setting. It requires more detailed and sophisticated analysis.



**Experience Assessment:**

I have published one or two papers in this area.

**Review Assessment: Checking Correctness Of Derivations And Theory:**

I carefully checked the derivations and theory.

**Review Assessment: Checking Correctness Of Experiments:**

I carefully checked the experiments.

**Review Assessment: Thoroughness In Paper Reading:**

I read the paper at least twice and used my best judgement in assessing the paper.

---

> ### Author Response · Authors · 2019-11-12
> **Response to R1**
>
> The baselines do not employ CL techniques since the main motivation behind the baseline experiments was to assess the impact and extent of catastrophic forgetting in RNNs, which we believed to be a necessary first step to highlight continual learning issues in the context of sequential data processing (being the literature on this topic still in its infancy, differently from what occurs with feedforward networks and machine vision applications). Results show that, unsurprisingly, the standard models are severely affected by catastrophic forgetting, supporting our claim for novel architectures and approaches addressing the issue for sequential models. It is within this context, that we introduce GIM as a possible solution to the issue: of course, there can be others and based on different approaches, e.g. an adaptation of elastic weight consolidation. Nevertheless, to the extent of our knowledge, there is, yet, no work in literature tackling catastrophic forgetting in continual learning for sequential problems.
> Nevertheless, we are taking in the reviewer suggestion and we will expand the analysis by adapting standard CL techniques to the recurrent scenario, assessing their performance in this novel context.
> We will provide ablation studies with respect to Augmented architectures and GIM architectures in future versions.

---

### Decision · Program_Chairs · 2019-12-19

**Decision:**

Reject

**Comment:**

This manuscript describes a continual learning approach where individual instances consist of sequences, such as language modeling. The paper consists of a definition of a problem setting, tasks in that problem setting, baselines (not based on existing continual learning approaches, which the authors argue is to highlight the need for such techniques, but with which the reviewers took issue), and a novel architecture.

Reviews focused on the gravity of the contribution. R1 and R2, in particular, argued that the paper is written as though the problem/benchmark definition is the main contribution. R2 mentions that in spite of this, the methods section jumps directly into the candidate architecture. As mentioned above, several reviewers also took issue with the fact that existing CL techniques are not employed as baselines. The authors engaged with reviewers and promised updates, but did not take the opportunity to update their paper.

As many of the reviewers' comments remain unaddressed and the authors' updates did not materialize, I recommend rejection, and encourage the authors to incorporate the feedback they have received in a future submission.